# CLASSICAL AND QUANTUM ALGORITHMS FOR ORTHOGONAL NEURAL NETWORKS

## ABSTRACT

Orthogonal neural networks have recently been introduced as a new type of neural networks imposing orthogonality on the weight matrices. They could achieve higher accuracy and avoid evanescent or explosive gradients for deep architectures. Several classical gradient descent methods have been proposed to preserve orthogonality while updating the weight matrices, but these techniques suffer from long running times and/or provide only approximate orthogonality. In this paper, we introduce a new type of neural network layer called Pyramidal Circuit, which implements an orthogonal matrix multiplication. It allows for gradient descent with perfect orthogonality with the same asymptotic running time as a standard fully connected layer. This algorithm is inspired by quantum computing and can therefore be applied on a classical computer as well as on a near term quantum computer. It could become the building block for quantum neural networks and faster orthogonal neural networks.

## 1 INTRODUCTION

In the evolution of neural network structures, adding constraints to the weight matrices has often been an effective path. Recently, orthogonal neural networks (OrthoNNs) have been proposed Jia et al. (2019); Wang et al. (2020); Nosarzewski (2018); Bansal et al. (2018) as a new type of neural networks for which, at each layer, the weight matrix should remain orthogonal. This property is useful to reach higher accuracy performance and avoid vanishing or exploding gradient for deep architectures. Several classical gradient descent methods have been proposed to preserve the orthogonality while updating the weight matrices. However, these techniques suffer from longer running time and sometimes only approximate the orthogonality. In particular, the main method for achieving orthogonality during training is to first perform the usual gradient descent to update the weight matrix (which is now not going to be orthogonal) and then perform Singular Value Decomposition to orthogonalize or almost orthogonalize the weight matrix. We can see then why achieving orthogonality hinders a fast training, since at every step an SVD computation needs to be performed (See Section A.1).

In the emergent field of quantum machine learning, several proposals have been made to implement neural networks. Some algorithms rely on long term and perfect quantum computers Kerenidis et al. (2020); Allcock et al. (2018), while others try to harness the existing quantum devices using variational circuits Cong et al. (2019); Farhi & Neven (2018). As in classical neural networks, they use gradient descent methods to update the quantum parameters of the circuits. Such quantum neural networks have been trained for very small sizes, however there is still a need to understand how such architectures will scale and whether they will provide efficient and accurate training.

In this work, we present a new training method for neural networks that preserves perfect orthogonality while having the same running time as usual gradient descent methods without the orthogonality condition, thus achieving the best of both worlds, most efficient training and perfect orthogonality.

The main idea comes from the quantum world, where we know that any quantum circuit corresponds to an operation described by a unitary matrix, which if we only use gates with real amplitudes is an orthogonal matrix. In particular, we propose a novel special-architecture quantum circuit, for which there is an efficient way to map the elements of the orthogonal weight matrix to the parameters of the gates of the quantum circuit and vice versa. In other words, while performing a gradient descent on

the elements of the weight matrix individually does not preserve orthogonality, performing a gradient descent on the parameters of the quantum circuit preserves orthogonality (since any quantum circuit with real parameters corresponds to an orthogonal matrix) and is equivalent to updating the weight matrix. We also prove that performing gradient descent on the parameters of the quantum circuit can be done efficiently classically (with constant update cost per parameter) thus concluding that there exists a quantum-inspired, but fully classical way of efficiently training perfectly orthogonal neural networks.

Moreover, the special-architecture quantum circuit we defined has many properties that make it a good candidate for NISQ implementation: it uses only one type of quantum gates, requires simple connectivity between the qubits, has depth linear in the input and output node sizes, and benefits from powerful error mitigation techniques that make it resilient to noise. This allows us to also propose an inference method running the quantum circuit on data which might offer a faster running time, given the shallow depth of the quantum circuit.

Our main contributions are summarized in Table 1, where we have considered the time to perform a feedforward pass, or one gradient descent step. A single neural network layer is considered, with input and output of size $n$.

| Algorithm | Feedforward Pass | Weight Matrix Update |
|---|---|---|
| Quantum Pyramidal Circuit (This work) | $2n/\delta^2 = O(n/\delta^2)$ | $O(n^2/\delta^2)$ |
| Classical Pyramidal Circuit (This work) | $2n(n-1) = O(n^2)$ | $O(n^2)$ |
| Classical Approximated OrthoNN (SVB) | $n^2 = O(n^2)$ | $O(n^3)$ |
| Classical Strict OrthoNN (Stiefel Manifold) | $n^2 = O(n^2)$ | $O(n^3)$ |
| Standard Neural Network (non orthogonal) | $n^2 = O(n^2)$ | $O(n^2)$ |

Table 1: Running times summary. $n$ is the size of the input and output vectors, $\delta$ is the error parameter in the quantum implementation. See Appendix Section A.1 for details on related work on classical approximated and strict OrthoNN.

## 2 A PARAMETRIZED QUANTUM CIRCUIT FOR ORTHOGONAL NEURAL NETWORKS

In this section, we define a special-architecture parametrized quantum circuit that will be useful for performing training and inference on orthogonal neural networks. As we said, the training will be completely classical in the end, but the intuition of the new method comes from this quantum circuit, while the inference can happen both classically or by applying this quantum circuit. A basic introduction to quantum computing concepts necessary for this work is given in the Appendix (Section A.3).

### 2.1 THE $RBS$ GATE

The quantum circuit proposed in this work (see Fig.1), which implements a fully connected neural network layer with an orthogonal weight matrix, uses only one type of quantum gate, the Reconfigurable Beam Splitter ($RBS$) gate. This two-qubit gate is parametrizable with one angle $\theta \in [0, 2\pi]$. Its matrix representation is given as:

$$RBS(\theta) = \begin{pmatrix} 1 & 0 & 0 & 0 \\ 0 & \cos\theta & \sin\theta & 0 \\ 0 & -\sin\theta & \cos\theta & 0 \\ 0 & 0 & 0 & 1 \end{pmatrix} \quad RBS(\theta): \begin{cases} |01\rangle \mapsto \cos\theta\,|01\rangle - \sin\theta\,|10\rangle \\ |10\rangle \mapsto \sin\theta\,|01\rangle + \cos\theta\,|10\rangle \end{cases} \quad (1)$$

We can think of this gate as a rotation in the two-dimensional subspace spanned by the basis $\{|01\rangle, |10\rangle\}$, while it acts as the identity in the remaining subspace $\{|00\rangle, |11\rangle\}$. Or equivalently, starting with two qubits, one in the $|0\rangle$ state and the other one in the state $|1\rangle$, the qubits can be swapped or not in superposition. The qubit $|1\rangle$ stays on its wire with amplitude $\cos\theta$ or switches with the other qubit with amplitude $+\sin\theta$ if the new wire is below ($|10\rangle \mapsto |01\rangle$) or $-\sin\theta$ if the new wire is above ($|01\rangle \mapsto |10\rangle$). Note that in the two other cases ($|00\rangle$ and $|11\rangle$) the $RBS$ gate acts as identity.

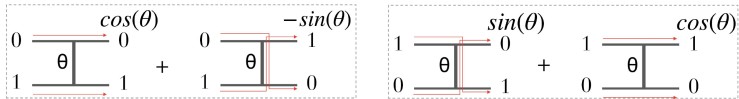

Figure 1: Representation of the quantum mapping from Eq.(1) on two qubits.

## 2.2 QUANTUM PYRAMIDAL CIRCUIT

We now propose a quantum circuit that implements an orthogonal layer of a neural network. The circuit is a pyramidal structure of $RBS$ gates, each with an independent angle, as represented in Fig.2a. In Section 2.3 and 3, more details are provided concerning respectively the input loading, and the equivalence with a neural network's orthogonal layer.

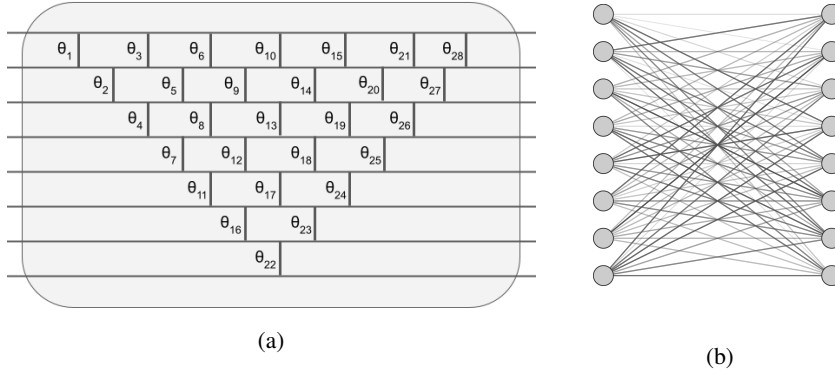

(a)

(b)

Figure 2: (a) Quantum circuit for an 8x8 fully connected, orthogonal layer. Each vertical line corresponds to an $RBS$ gate with its angle parameter. And (b), the equivalent classical orthogonal neural network 8x8 layer.

To mimic a given classical layer with a quantum circuit, the number of output qubits should be the size of the classical layer's output. We refer to the *square case* when the input and output sizes are equal, and to the *rectangular case* otherwise (Fig.3a).

The important property to note is that the number of parameters of the quantum pyramidal circuit corresponding to a neural network layer of size $n \times d$ is $(2n - 1 - d) * d/2$ exactly the same as the number of degrees of freedom of an orthogonal matrix of dimension $n \times d$.

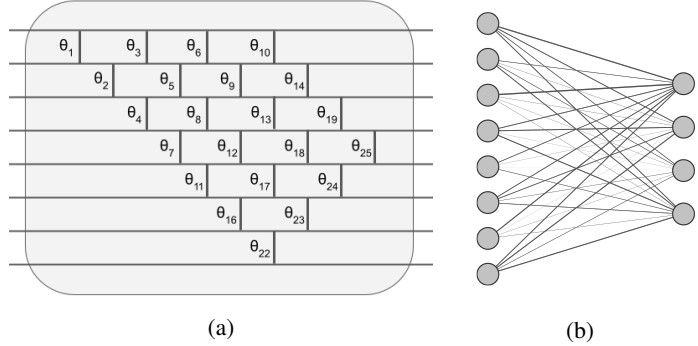

(a)

(b)

Figure 3: (a) Quantum circuit for a *rectangular* 8x4 fully connected orthogonal layer, and (b) the equivalent 8x4 classical orthogonal neural network. They both have 22 free parameters.

For simplicity, we pursue our analysis using only the *square case* but everything can be easily extended to the rectangular case. As we said, the full pyramidal structure of the quantum circuit

described above imposes the number of free parameters to be $N = n(n-1)/2$, the exact number of free parameters to specify a $n \times n$ orthogonal matrix.

In Section 3 we will show how the parameters of the gates of this pyramidal circuit can be easily related to the elements of the orthogonal matrix of size $n \times n$ that describes it. We note that alternative architectures can be imagined as long as the number of gate parameters is equal to the parameters of the orthogonal weight matrix and a simple mapping between them and the elements of the weight matrix can be found.

Note finally that this circuit has linear depth and is convenient for near term quantum hardware platforms with restricted connectivity. Indeed, the distribution of the $RBS$ gates requires only nearest neighbor connectivity between qubits.

### 2.3 LOADING THE DATA

Before applying the quantum pyramidal circuit, we will need to upload the classical data into the quantum circuit. We will use one qubit per feature of the data. For this, we use a unary amplitude encoding of the input data. Let's consider an input sample $x = (x_0, \cdots, x_{n-1}) \in \mathbb{R}^n$, such that $\|x\|_2 = 1$. We will encode it in a superposition of unary states:

$$|x\rangle = x_0 |10\cdots0\rangle + x_1 |010\cdots0\rangle + \cdots + x_{n-1} |0\cdots01\rangle \tag{2}$$

We can also rewrite the previous state as $|x\rangle = \sum_{i=0}^{n-1} x_i |e_i\rangle$, where $|e_i\rangle$ represents the $i^{th}$ unary state with a $|1\rangle$ in the $i^{th}$ position $|0\cdots010\cdots0\rangle$. Recent work Kerenidis (U.S. Patent Application No. 16/986,553 and 16/987,235, 2020) proposed a logarithmic depth data loader circuit for loading such states. Here we will use a much simpler circuit. It is a linear depth cascade of $n-1$ $RBS$ gates which, due to the particular structure of our quantum pyramidal circuit, only adds 2 extra steps to our circuit.

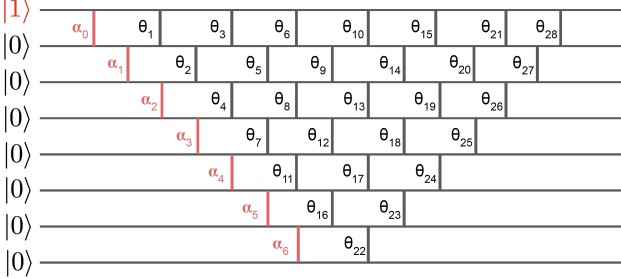

Figure 4: The 8 dimensional linear data loader circuit (in red) is efficiently embedded before the pyramidal circuit. The input state is the first unary state. The angles parameters $\alpha_0, \cdots, \alpha_{n-2}$ are classically pre-computed from the input vector.

The circuit starts in the all $|0\rangle$ state and flips the first qubit using an $X$ gate, in order to obtain the unary state $|10\cdots0\rangle$ as shown in Fig.4. Then a cascade of $RBS$ gates allow to create the state $|x\rangle$ using a set of $n-1$ angles $\alpha_0, \cdots, \alpha_{n-2}$. Using Eq.(1), we will choose the angles such that, after the first $RBS$ gate of the loader, the qubits would be in the state $x_0 |100\cdots\rangle + \sin(\alpha_0) |010\cdots\rangle$ and after the second one in the state $x_0 |100\cdots\rangle + x_1 |010\cdots\rangle + \sin(\alpha_0)\sin(\alpha_1) |001\cdots\rangle$ and so on, until obtaining $|x\rangle$ as in Eq.(2). To this end, we simply perform a classical preprocessing to compute recursively the $n-1$ loading angles, in time $O(n)$. We choose $\alpha_0 = \arccos(x_0)$, $\alpha_1 = \arccos(x_1 \sin^{-1}(\alpha_0))$, $\alpha_2 = \arccos(x_2 \sin^{-1}(\alpha_0)\sin^{-1}(\alpha_1))$ and so on.

The ability of loading data in such a way relies on the assumption that each input vector is normalized, i.e. $\|x\|_2 = 1$. This normalization constraint could seem arbitrary and impact the ability to learn from the data. In fact, in the case of an orthogonal neural network, this normalization shouldn't degrade the training because orthogonal weight matrices are in fact orthonormal and thus norm-preserving. Hence, changing the norm of the input vector, by diving each component by $\|x\|_2$, in both classical and quantum settings is not a problem. The normalization would impose that each input has the same norm, or the same "luminosity" in the context of images, which can be helpful or harmful depending on the case.

## 3  ORTHONNS FEEDFORWARD PASS

In this section, we will detail the effect of the quantum pyramidal circuit on an input encoded in a unary basis, as in Eq.(2). We will also see in the end how to simulate this quantum circuit classically with a small overhead and thus be able to provide a fully classical scheme.

Let's first consider one pure unary input, where only the qubit $j$ is in state $|1\rangle$ (e.g. $|00000010\rangle$). This unary input will be transformed into a superposition of unary states, each with an amplitude. If we consider again only one of these possible unary outputs, where only the qubit $i$ is in state $|1\rangle$, its amplitude can be interpreted as a conditional amplitude to transfer the $|1\rangle$ from qubit $j$ to qubit $i$. Intuitively, this value is the sum of the quantum amplitudes associated to each possible path that *connects* the qubit $j$ to qubit $i$, as shown in Fig.5. Using this image of *connectivity* between input and output qubits, we can construct a matrix $W \in \mathbb{R}^{n \times n}$, where each element $W_{ij}$ is the overall conditional amplitude to transfer the $|1\rangle$ from qubit $j$ to qubit $i$.

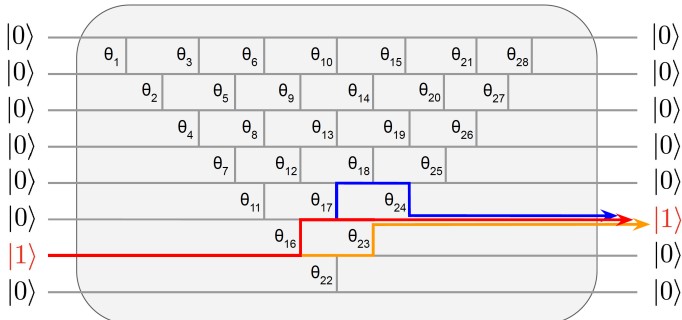

Figure 5: The three possibles paths from the $7^{th}$ unary state to the $6^{th}$ unary state, on an 8x8 quantum pyramidal circuit.

Fig.5 shows an example where exactly three paths can be taken to map the input qubit $j = 6$ (the $7^{th}$ unary state) to the qubit $i = 5$ (the $6^{th}$ unary state). Each path comes with a certain amplitude. For instance, one of the paths (the red one in Fig.5) moves up at the first gate, and then stays put in the next three gates, with a resulting amplitude of $-\sin(\theta_{16})\cos(\theta_{17})\cos(\theta_{23})\cos(\theta_{24})$. The sum of the amplitudes of all possible paths give us the element $W_{56}$ of the matrix $W$ (where, for simplicity, $s(\theta)$ and $c(\theta)$ respectively stand for $\sin(\theta)$ and $\cos(\theta)$):

$$W_{56} = -s(\theta_{16})c(\theta_{22})s(\theta_{23}) - s(\theta_{16})c(\theta_{17})c(\theta_{23})c(\theta_{24}) + s(\theta_{16})s(\theta_{17})c(\theta_{18})s(\theta_{24}) \quad (3)$$

In fact, the $n \times n$ matrix $W$ can be seen as the unitary matrix of our quantum circuit if we solely consider the unary basis, which is specified by the parameters of the quantum gates. A unitary is a complex unitary matrix, but in our case, with only real operations, the matrix is simply orthogonal. This proves the correspondence between any matrix $W$ and the pyramidal quantum circuit.

The full unitary $U_W$ in the Hilbert Space of our $n$-qubit quantum circuit is a $2^n \times 2^n$ matrix with the $n \times n$ matrix $W$ embedded in it as a submatrix on the unary basis. This is achieved by loading the data as unary states and by using only $RBS$ gates that keep the number of 0s and 1s constant.

For instance, as shown in Fig.6, a 3-qubit pyramidal circuit is described as a unique $3 \times 3$ matrix, that can be easily verified to be orthogonal.

In Fig.5, we considered the case of single unary for both the input and output. But with actual data, as seen in Section 2.3, input and output states are in fact a superposition of unary states. Thanks to the linearity of quantum mechanics in absence of measurements, the previous descriptions remain valid and can be applied on a linear combination of unary states.

Let's consider an input vector $x \in \mathbb{R}^n$ encoded as a quantum state $|x\rangle = \sum_{i=0}^{n-1} x_i |e_i\rangle$ where $|e_i\rangle$ represents the $i^{th}$ unary state (see Section 2.3). By definition of $W$, each unary $|e_i\rangle$ will undergo a proper evolution $|e_i\rangle \mapsto \sum_{j=0}^{n-1} W_{ij} |e_j\rangle$. This yields, by linearity, to the following mapping

$$|x\rangle \mapsto \sum_{i,j} W_{ij} x_i |e_j\rangle \quad (4)$$

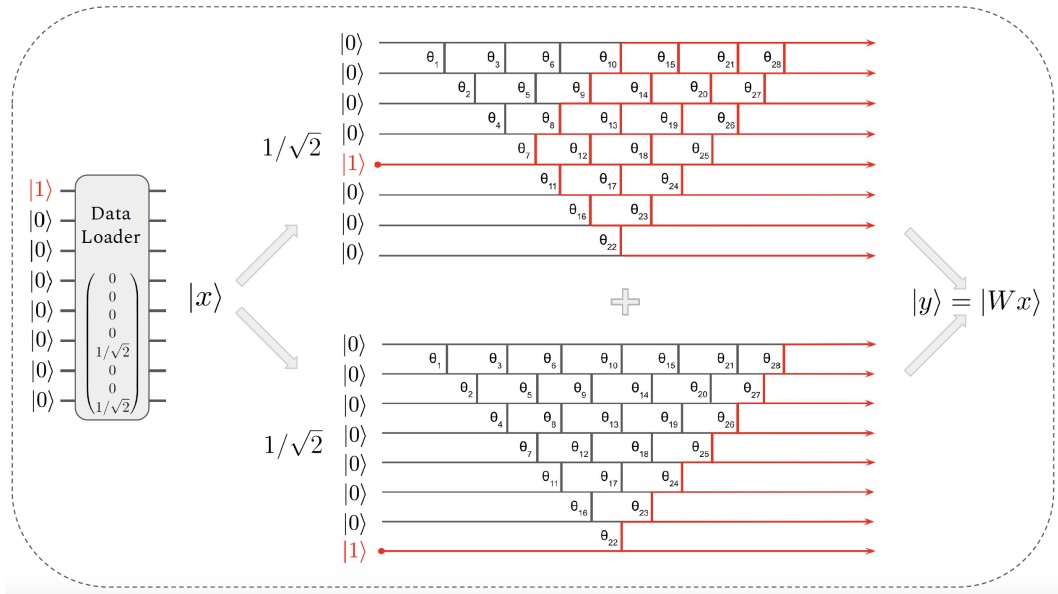

Figure 6: Example of a 3 qubits pyramidal circuit and the equivalent orthogonal matrix. $c(\theta)$ and $s(\theta)$ respectively stand for $\cos(\theta)$ and $\sin(\theta)$.

Figure 7: Schematic representation of a pyramidal circuit applied on a loaded vector $x$ with two non-zero values. The output is the unary encoding of $y = Wx$ where $W$ is the corresponding orthogonal matrix associated with the circuit.

As explained above, our quantum circuit is equivalently described by the sparse unitary $U_W \in \mathbb{R}^{2^n \times 2^n}$ or on the unary basis by the matrix $W \in \mathbb{R}^{n \times n}$. This can be summarized with

$$U_W \left| x \right\rangle = \left| Wx \right\rangle \tag{5}$$

We see from Eq.(4) and Eq.(5) that the output is in fact $|y\rangle$, the unary encoding of the vector $y = Wx$, which is the output of a matrix multiplication between the $n \times n$ orthogonal matrix $W$ and the input $x \in \mathbb{R}^n$. As expected, each element of $y$ is given by $y_k = \sum_{i=0}^{n-1} W_{ik} x_i$. See Fig.7 for a diagram representation of this mapping.

Therefore, for any given neural network's orthogonal layer, there is a quantum pyramidal circuit that reproduces it. On the other hand, any quantum pyramidal circuit is implementing an orthogonal layer of some sort. Additional details concerning multi-layers branching, the tomography at the end of each layer, and the way to apply the non linearities are given in the Appendix, Section A.4.

As a side note, we can ask if a circuit with only $\log(n)$ qubits could also implement an orthogonal matrix multiplication of size $n \times n$. Indeed, it would be a unitary matrix in $\mathbb{R}^{n \times n}$, but since the circuit should also have $n(n-1)/2$ free parameters to tune, this would come at a cost of large depth, potentially unsuitable for NISQ devices.

**Classical implementation** While we presented the quantum pyramidal circuit as the inspiration of the new methods for orthogonal neural networks, it is not hard to see that these quantum circuits can be simulated classical with a small overhead, thus yielding classical methods for orthogonal neural networks.

This classical algorithm is simply the simulation of the quantum pyramidal circuit, where each $RBS$ gate is replaced by a planar rotation between its two inputs.

As shown in Fig.8, we propose a similar classical pyramidal circuit, where each layer is constituted of $\frac{n(n-1)}{2}$ planar rotations, for a total of $4 \times \frac{n(n-1)}{2} = O(n^2)$ basic operations. Therefore our single layer feedforward pass has the same complexity $O(n^2)$ as the usual matrix multiplication.

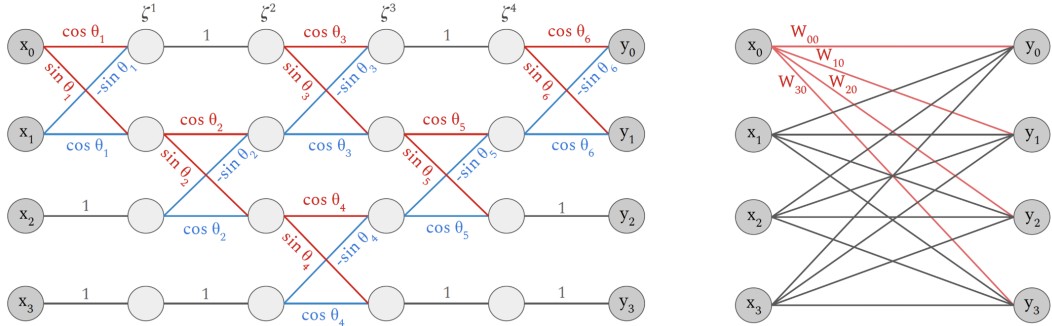

Figure 8: Classical representation of a single orthogonal layer on a 4x4 case ($n$=4) performing $x \mapsto y = Wx$. The angles and the weights can be chosen such that our classical pyramidal circuit (left) and normal classical network (right) are equivalent. Each connecting line represents a scalar multiplication with the value indicated. On the classical pyramidal circuit (left), *inner layers* $\zeta^\lambda$ are displayed. A *timestep* corresponds to the lines in between two *inner layers* (see Section 4 for definitions).

One may still have an advantage in performing the quantum circuit for inference, since the quantum circuit has depth $O(n)$, instead of the $O(n^2)$ classical complexity of the matrix-vector multiplication. Nevertheless, as we will see below, the main advantage of our method is that we can now train orthogonal weight matrices classically in time $O(n^2)$, instead of the previously best-known $O(n^3)$.

# 4 ORTHONN TRAINING: ANGLE'S GRADIENT CALCULATION AND ORTHOGONAL MATRIX UPDATE

Basic introduction and notations to the backpropagation in fully connected neural networks are given in the Appendix, Section A.2.

Looking through the prism of our pyramidal quantum circuit, the parameters to update are no longer the individual elements of the weight matrices directly, but the angles of the RBS gates that give rise to these matrices. Thus, we need to design an adaptation of the backpropagation method to our setting based on the angles. We will start by introducing some notation for a single layer $\ell$, which will not be explicit in the notation for simplicity. We assume we have as many output bits as input bits, but this can easily be extended to the *rectangular* case.

We first introduce the notion of *timesteps* inside each layer, which correspond to the computational steps in the pyramidal structure of the circuit (see Fig.9). It is easy to show that for $n$ inputs, there will be $2n - 3$ such *timesteps*, each one indexed by an integer $\lambda \in [0, \cdots, \lambda_{max}]$. Applying a timestep consists in applying the matrix $w^\lambda$, made of all the RBS gates aligned vertically at this timestep ($w^\lambda$ is the unitary in the unary basis, see Section 3 for details). Each time a timestep is applied, the resulting state is a vector in the unary basis named *inner layer* and noted $\zeta^\lambda$. This evolution can be written as $\zeta^{\lambda+1} = w^\lambda \cdot \zeta^\lambda$. We use this notation similar to the real layer $\ell$, with the weight matrix $W^\ell$ and the resulting vector $z^\ell$ (see Section A.2).

In fact we have the correspondences $\zeta^0 = a^{\ell-1}$ for the first *inner layer*, which is the input of the actual layer, and $z^\ell = w^{\lambda_{max}} \cdot \zeta^{\lambda_{max}}$ for the last one. We also have $W^\ell = w^{\lambda_{max}} \cdots w^1 w^0$. We use the same kind of notation for the backpropagation errors. At each timestep $\lambda$ we define an *inner error* $\delta^\lambda = \frac{\partial C}{\partial \zeta^\lambda}$. This definition is similar to the layer error $\Delta^\ell = \frac{\partial C}{\partial z^\ell}$. In fact we will use the same backpropagation formulas, without non linearities, to retrieve each *inner error*

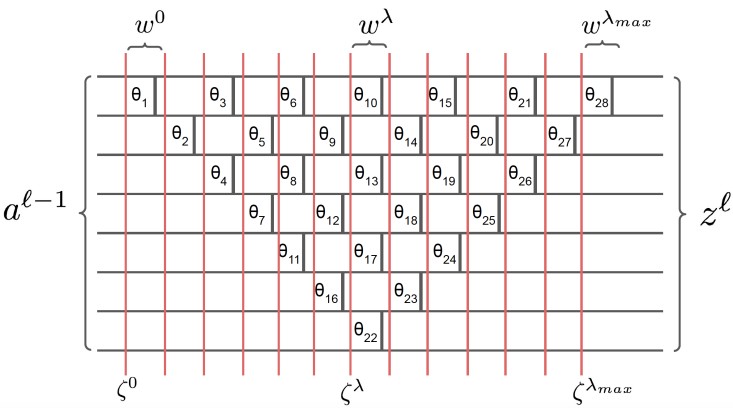

Figure 9: Quantum circuit for one neural network layer divided into *timesteps* (red vertical lines) $\lambda \in [0, \cdots, \lambda_{max}]$. Each timestep corresponds to an *inner layer* $\zeta^\lambda$ and an *inner error* $\delta^\lambda$. The part of the circuit between two timesteps is an unitary matrix $w^\lambda$ in the unary basis.

vector $\delta^\lambda = (w^\lambda)^T \cdot \delta^{\lambda+1}$. In particular, for the last timestep, the first to be calculated, we have $\delta^{\lambda_{max}} = (w_{max}^\lambda)^T \cdot \Delta^\ell$. Finally, we can retrieve the error at the previous layer $\ell - 1$ using the correspondence $\Delta^{\ell-1} = \delta^0 \odot \sigma'(z^\ell)$.

The reason for this breakdown into timesteps is the ability to efficiently obtain the gradient with respect to each angle. Let's consider one gate with angle $\theta_i$, acting at the timestep $\lambda$ on qubits $i$ and $i + 1$. We will decompose the gradient $\frac{\partial \mathcal{C}}{\partial \theta_i}$ using each component, indexed by the integer $k$, of the *inner layer* and *inner error* vectors $\frac{\partial \mathcal{C}}{\partial \theta_i} = \sum_k \frac{\partial \mathcal{C}}{\partial \zeta_k^{\lambda+1}} \frac{\partial \zeta_k^{\lambda+1}}{\partial \theta_i} = \sum_k \delta_k^{\lambda+1} \frac{\partial (w_k^\lambda \cdot \zeta^\lambda)}{\partial \theta_i}$.

Since timestep $\lambda$ is only composed of separated RBS gates, the matrix $w^\lambda$ consists of diagonally arranged 2x2 block submatrices given in Eq.(1). Only one of these submatrices depends on the angle $\theta$ considered here, at the position $i$ and $i + 1$ in the matrix. The above gradient can be rewritten as:

$$\frac{\partial \mathcal{C}}{\partial \theta_i} = \delta_i^{\lambda+1}(-\sin(\theta_i)\zeta_i^\lambda + \cos(\theta_i)\zeta_{i+1}^\lambda) + \delta_{i+1}^{\lambda+1}(-\cos(\theta_i)\zeta_i^\lambda - \sin(\theta_i)\zeta_{i+1}^\lambda) \tag{6}$$

Therefore we have shown a way to compute each angle gradient: During the feedforward pass, one must apply sequentially each of the $2n - 3 = O(n)$ timesteps, and store the resulting vectors, the *inner layers* $\zeta^\lambda$. During the backpropagation, one obtains the *inner errors* $\delta^\lambda$ by applying the timesteps in reverse. One can finally use a gradient descent on each angle $\theta_i$, while preserving the orthogonality of the overall equivalent weight matrix $\theta_i^\ell \leftarrow \theta_i^\ell - \lambda \frac{\partial \mathcal{C}}{\partial \theta_i^\ell}$.

An interesting aspect of this gradient descent is the fact that the optimization is performed in the angle landscape, and not on the equivalent weight landscape. These landscapes can potentially be different and hence our optimization can produce different models. We leave open the question of finding a theoretical argument to compare the properties of both landscapes.

As one can see from the above description, this is in fact a classical algorithm to obtain the angle's gradients, which allows us to train our OrthoNN efficiently classically while preserving the strict orthogonality. To obtain the angle's gradient, one needs to store the $2n - 3$ *inner layers* $\zeta^\lambda$ during the feedforward pass. Next, given the error at the following layer, we perform a backward loop on each *timestep* (see Fig.8). At each *timestep*, we obtain the gradient for each angle parameter, by simply applying Eq.(4). This requires $O(1)$ operations for each angle. Since there are at most $n/2$ angles per *timesteps*, estimating gradients has a complexity of $O(n^2)$. After each *timestep*, the next *inner error* $\delta^{\lambda-1}$ is computed as well, using at most $4n/2$ operations.

In the end, our classical algorithm allows us to compute the gradients of the $n(n - 1)/2$ angles in $O(n^2)$, in order to perform a gradient descent respecting the strict orthogonality of the weight matrix. This is considerably faster than previous methods based on Singular Value Decomposition methods and provides a training method that is asymptotically as fast as for normal neural networks, while providing the extra property of orthogonality.

## 5 NUMERICAL EXPERIMENTS

We performed basic numerical experiments to verify the abilities of our pyramidal circuit, on the standard MNIST dataset LeCun & Cortes (2010). Note that current quantum hardware and software are not yet suited for bigger experiments. We first compared the training of our Classical OrthoNN to the SVB algorithm from Jia et al. (2019) (see Section A.1).Results as reported in Fig.10, and more in the Appendix, Section A.4.3. These small scale tests confirmed that the pyramidal circuits and the corresponding gradient descent on the angles were efficient for learning a classification task.

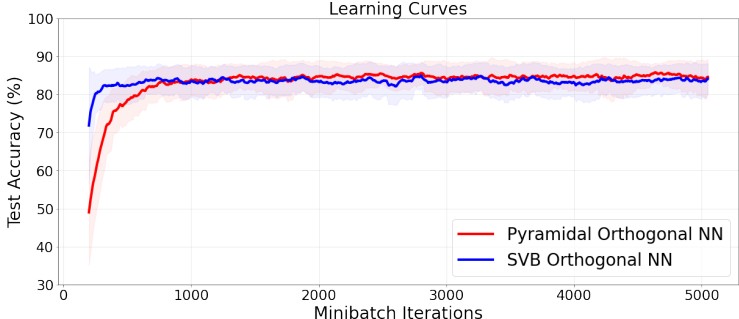

Figure 10: Training comparison between a [16,8,4] SVB OrthoNN from Jia et al. (2019) and our classical pyramidal OrthoNN. Test accuracy on 1000 samples during 50 epochs of training on the MNIST dataset on 5000 samples. Initial dimensionality reduction (PCA) was on the samples to fit the input layer of the networks. Shaded areas indicate the accuracy variance during minibatch updates of size 50.

Then, we implemented the quantum circuit on a real quantum computer provided by IBM. We used a 16 and 5 qubits device to perform respectively a [8,2] and a [4,2] orthogonal layer. A pyramidal OrthoNN was trained classically, and the resulting angles were transferred to test the quantum circuit on a classification task on classes 6 and 9 of the MNIST dataset, over 500 samples. We compared the real experiment with a simulated one, and the classical pyramidal circuit as well. Results are reported in Table 2.

| Platform | Accuracy | Platform | Accuracy |
|---|---|---|---|
| Classical circuit [4,2] | 98,4% | Classical circuit [8,2] | 97,4% |
| IBM Simulator [4,2] | 98,4% | IBM Simulator [8,2] | 97,4% |
| *ibmq_bogota v1.4.32* [4,2] | 98,0% | *ibmq_guadalupe v1.2.17* [8,2] | 95,0% |

Table 2: Results of the Pyramidal OrthoNN on real quantum computers. *ibmq_bogota v1.4.32* and *ibmq_guadalupe v1.2.17* are respectively 5 and 16 qubits quantum computers.

## 6 CONCLUSION AND OUTLOOK

In this work, we have proposed for the first time training methods for orthogonal neural networks (OrthoNNs) that run in quadratic time, a significant improvement from previous methods based on Singular Value Decomposition. The main idea of our method is to replace usual weights and orthogonal matrices with an equivalent pyramidal circuit made of two-dimensional rotations. Each rotation is parametrizable by an angle, and the gradient descent takes place in the angle's optimization landscape. This unique type of gradient backpropagation ensures perfect orthogonality of the weight matrices while improving the running time compared to previous works. Moreover, we propose both classical and quantum methods for inference, where the forward pass on a near term quantum computer would provide a provable advantage in the running time. This work expands the field of quantum deep learning by introducing new tools, concepts, and equivalences with classical deep learning theory. We have highlighted open questions regarding the construction of such pyramidal circuits for neural networks and their potential new advantages in terms of execution time, accuracy, and learning properties.

## 7 Reproducibility Statement

In this paper, we have introduced new methods for implementing and training orthogonal neural networks. We have explained in detail each part of both algorithms' implementation (see Appendix for more details). The dataset used here for basic simulation is the well-known MNIST dataset, easily accessible to everyone.

It is in our belief that anyone with classical software skills can re-implement the classical algorithm (see Fig.9), in Python for instance. Similarly, on any quantum software platform, one can implement the quantum gates given in Fig.11 and perform the whole algorithm using Fig.12, Fig.10, and mathematical details through the paper. This would allow simulating the quantum circuit using accessible platforms such as IBM's qiskit.

However, in order to perform the actual experiment on a quantum computer currently accessible, as we have done on IBM 5 and 16 qubits devices, one must have access to their service (or any other provider). Some quantum computers are available for free, but others require specific agreements.

## 8 Ethics Statement

Our work on quantum and classical algorithms for orthogonal neural networks is mainly a theoretical work aiming at improving the complexity of neural networks training on current computers and near term quantum computers. Therefore, any ethical concerns regarding classical neural networks may also concern our work.

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

# A APPENDIX

## A.1 RELATED WORK ON CLASSICAL ORTHONNS

The idea behind Orthogonal Neural Networks (OrthoNNs) is to add a constraint to the weight matrices corresponding to the layers of a neural network. Imposing orthogonality to these matrices has theoretical and practical benefits in the generalization error Jia et al. (2019). Orthogonality ensures a low weights redundancy and preserves the magnitude of the weight matrix's eigenvalues to avoid vanishing gradients. In terms of complexity, for a single layer, the feedforward pass of an OrthoNN is simply a matrix multiplication, hence has a running time of $O(n^2)$ if $n \times n$ is the size of the orthogonal matrix. It is also interesting to note that OrthoNNs have been generalized to Convolutional Neural Networks Wang et al. (2020).

The main difficulty of OrthoNNs is to preserve the orthogonality of the matrices while updating them during gradient descent. Several algorithms have been proposed to this end Wang et al. (2020); Bansal et al. (2018); Lezcano-Casado & Martınez-Rubio (2019), but they all point that pure orthogonality is computationally hard to conserve. Therefore, previous works allow for approximations: strict orthogonality is no longer required, and the matrices are often pushed toward orthogonality using regularization techniques during weights update.

We present two algorithms from Jia et al. (2019) for updating orthogonal matrices.

The first algorithm is an approximated one, called *Singular Value Bounding* (SVB). It starts by applying the usual gradient descent update on the matrix, therefore making it not orthogonal anymore. Then, the singular values of the new matrix are extracted using Singular Value Decomposition (SVD), their values are manually pushed to be close to 1, and the matrix is recomposed hence enforcing orthogonality. This method shows less advantage on practical experiments Jia et al. (2019). It has a complexity of $O(n^3)$ due to the SVD, which in practice is better than the next algorithm. Note that this running time is still longer than $O(n^2)$, the running time to perform standard gradient descent.

The second algorithm can be considered perfect since it ensures strict orthogonality by performing the gradient descent in the manifold of orthogonal matrices, called the Stiefel Manifold. In practice Jia et al. (2019) this method showed a substantially advantageous classification results on standard datasets. This algorithm requires $O(n^3)$ operations, but is very prohibitive in practice. We give an informal step-by-step detail of this algorithm (see Jia et al. (2019), Appendix G for details):

1. Compute the gradient $G$ of the weight matrix $W$.
2. Project the gradient matrix $G$ in the tangent space, (The space tangent to the manifold at this point $W$): multiply $G$ by some other matrices based on $W$:

$$(I - WW^T)G + \frac{1}{2}W(W^T G - G^T W)$$

   This requires several matrix-matrix multiplications. In the case of square $n \times n$ matrices, each has complexity $O(n^3)$. the result of this projection is called the *manifold gradient* $\Omega$.
3. update $W' = W - \eta\Omega$, where $\eta$ is the chosen learning rate.
4. Perform a *retraction* from the tangent space to the manifold. To do so we multiply $W'$ by $Q$ factor of the *QR decomposition*, obtained using Gram Schmidt orthonormalization, which has complexity $O(2n^3)$.

## A.2 CLASSICAL BACKPROPAGATION ALGORITHM

The backpropagation in a fully connected neural network is a well know and efficient procedure to update the weight matrix at each layer Hecht-Nielsen (1992); Rojas (1996). At layer $\ell$, we note its weight matrices $W^\ell$ and biases $b^\ell$. Each layer is followed by a non linear function $\sigma$, and can therefore be written as

$$a^\ell = \sigma(W^\ell \cdot a^{\ell-1} + b^\ell) = \sigma(z^\ell) \tag{7}$$

After the last layer, one can define a cost function $\mathcal{C}$ that compares the output to the ground truth. The goal is to calculate the gradient of $\mathcal{C}$ with respect to each weight and bias, namely $\frac{\partial \mathcal{C}}{\partial W^\ell}$ and

$\frac{\partial \mathcal{C}}{\partial b^\ell}$. In the backpropagation, we start by calculating these gradients for the last layer, then propagate back to the first layer.

We will require to obtain the *error* vector at layer $\ell$ defined by $\Delta^\ell = \frac{\partial \mathcal{C}}{\partial z^\ell}$. One can show the backward recursive relation $\Delta^\ell = (W^{\ell+1})^T \cdot \Delta^{\ell+1} \odot \sigma'(z^\ell)$, where $\odot$ symbolizes the Hadamard product, or entry-wise multiplication. Note that the previous computation requires simply to apply the layer (*ie* apply matrix multiplication) in reverse. We can then show that each element of the weight gradient matrix at layer $\ell$ is given by $\frac{\partial \mathcal{C}}{\partial W_{jk}^\ell} = \Delta_j^\ell \cdot a_1^{\ell-1}$. Similarly, the gradient with respect to the biases is easily defined as $\frac{\partial \mathcal{C}}{\partial b_j^\ell} = \Delta_j^\ell$.

Once these gradients are computed, we update the parameters using the gradient descent rule, with learning rate $\lambda$:

$$W_{jk}^\ell \leftarrow W_{jk}^\ell - \lambda \frac{\partial \mathcal{C}}{\partial W_{jk}^\ell} \quad ; \quad b_j^\ell \leftarrow b_j^\ell - \lambda \frac{\partial \mathcal{C}}{\partial b_j^\ell} \tag{8}$$

### A.3 Preliminaries in Quantum Computing

We present a succinct broad-audience quantum information background necessary for this work. See Nielsen & Chuang (2002) for a detailed course.

**Qubits:** In classical computing, a bit can be either 0 or 1. With a quantum information perspective, a quantum bit or *qubit* can be is state $|0\rangle$, $|1\rangle$. We use the *braket* notation $|\cdot\rangle$ to specify the quantum nature of the bit. The qubits can be in superposition of both states $\alpha |0\rangle + \beta |1\rangle$ where $\alpha, \beta \in \mathbb{C}$ such that $|\alpha|^2 + |\beta|^2 = 1$. The coefficients $\alpha$ and $\beta$ are called *amplitudes*. The probabilities of observing either 0 or 1 when *measuring* the qubit are linked to the amplitudes:

$$p(0) = |\alpha|^2, \quad p(1) = |\beta|^2 \tag{9}$$

As quantum physics teaches us, any superposition is possible before the measurement, which gives special abilities in terms of computation. With a $n$ qubits, $2^n$ possible binary combinations (e.g. $|01 \cdots 1001\rangle$) can exist simultaneously, each with its own amplitude.

A $n$ qubits system can be represented as a normalized vector in a $2^n$ dimensional Hilbert space. A multiple qubit system is called a quantum *register*. If $|p\rangle$ and $|q\rangle$ are two quantum states or quantum registers, the whole system can be represented as a tensor product $|p\rangle \otimes |q\rangle$, also written as $|p\rangle |q\rangle$ or $|p, q\rangle$.

**Quantum Computation:** As logical gates in classical circuits, qubits or quantum registers are processed using quantum gates. A quantum gate is a *unitary* mapping in the Hilbert space, preserving the unit norm of the quantum state vector. Therefore, a quantum gate acting on $n$ qubits is a matrix $U \in \mathbb{C}^{2^n}$ such that $UU^\dagger = U^\dagger U = I$, with $U^\dagger$ being the adjoint, or conjugate transpose, of $U$.

Common single qubit gates include the Hadamard gate $\frac{1}{\sqrt{2}} \begin{pmatrix} 1 & 1 \\ 1 & -1 \end{pmatrix}$ that maps $|0\rangle \mapsto \frac{1}{\sqrt{2}}(|0\rangle + |1\rangle)$ and $|1\rangle \mapsto \frac{1}{\sqrt{2}}(|0\rangle - |1\rangle)$, creating the quantum superposition, the NOT gate $\begin{pmatrix} 0 & 1 \\ 1 & 0 \end{pmatrix}$ that permutes $|0\rangle$ and $|1\rangle$, or $R_y$ rotation gate parametrized by an angle $\theta$, given by $\begin{pmatrix} \cos(\theta/2) & -\sin(\theta/2) \\ \sin(\theta/2) & \cos(\theta/2) \end{pmatrix}$.

Common two-qubits gates includes the CNOT gate $\begin{pmatrix} 1 & 0 & 0 & 0 \\ 0 & 1 & 0 & 0 \\ 0 & 0 & 0 & 1 \\ 0 & 0 & 1 & 0 \end{pmatrix}$ which is a NOT gate applied on the second qubit only if the first one is in state $|1\rangle$, or similarly the CZ gate $\begin{pmatrix} 1 & 0 & 0 & 0 \\ 0 & 1 & 0 & 0 \\ 0 & 0 & 1 & 0 \\ 0 & 0 & 0 & -1 \end{pmatrix}$.

In this work, we use the *RBS* gate given in Eq.(1). This gate can be implemented rather easily, either as a native gate, known as $FSIM$ Foxen et al. (2020), or using four Hadamard gates, two $R_y$ rotation gates, and two two-qubits CZ gates:

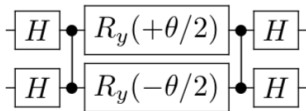

Figure 11: A possible decomposition of the $RBS(\theta)$ gate.

The main advantage of quantum gates is their ability to be applied to a superposition of inputs. Indeed, given a gate $U$ such that $U |x\rangle \mapsto |f(x)\rangle$, we can apply it to all possible combinations of $x$ at once $U(\frac{1}{C} \sum_x |x\rangle) \mapsto \frac{1}{C} \sum_x |f(x)\rangle$.

### A.4    Additional Details on the Quantum Pyramidal Circuit

#### A.4.1    Tomography and Error Mitigation

As shown in Fig.7, when using the quantum circuit, the output is a quantum state $|y\rangle = |Wx\rangle$. As often in quantum machine learning Aaronson (2015), it is important to go all to way and consider the cost of retrieving classical outputs, using a procedure called tomography. In our case this is even more crucial since, between each layer, the quantum output will be converted into a classical one in order to apply a non linear function, and then reloaded for the next layer.

**Error Mitigation**    Before detailing the tomography procedure, it is interesting to notice that with our restriction to unary states, a strong benefit appears for error mitigation purposes. Indeed, as we expect to obtain only quantum superposition of unary states at every layer, we can post process our measurements and discard the ones that present non unary states (*i.e.* states with more than one qubit in state $|1\rangle$, or the ground state). The most expected error is a bit flip between $|1\rangle$ and $|0\rangle$. The case where two bit flips happened, which would pass through our error mitigation, is even less probable.

**Tomography**    Retrieving the amplitudes of a quantum state comes at cost of multiple measurements. which requires running the circuit multiples times, hence adding a multiplicative overhead in the running time. A finite number of samples is also a source of approximation in the final result. In this work, we will allow for $\ell_\infty$ errors Kerenidis et al. (2020). The $\ell_\infty$ tomography on a quantum state $|y\rangle$ with unary encoding on $n$ qubits requires $O(\log(n)/\delta^2)$ measurements, where $\delta > 0$ is the error threshold allowed. For each $j \in [n]$, $|y_j|$ will be obtained with an absolute error $\delta$, and if $|y_j| < \delta$, it will most probably not be measured, hence set to 0. In practice, one would perform as many measurements as is convenient during the experiment, and deduce the equivalent precision $\delta$ from the number of measurements made.

The next logical part is to obtain the sign of each component of the vector. Indeed, we only measure probabilities that are the square module of the quantum amplitudes (see Section A.3). In the case of neural networks, it is important to obtain the sign of the layer's components in order to apply certain type of non linearities. For instance, the ReLu activation function is often used to set all negative components to 0.

In Fig.12, we propose a specific enhancement to our circuit to obtain the signs of the vector's components at low cost. The sign retrieval procedure consists of three parts.

a) The circuit is first applied as described above, allowing to retrieve each squared amplitude $y_j^2$ with precision $\delta > 0$ using the $\ell_\infty$ tomography. The probability of measuring the unary state $|e_1\rangle$ (*i.e.* $|100...\rangle$), is $p(e_1) = y_1^2$.

b) We apply the same steps a second time on a modified circuit. It has additional *RBS* gates with angle $\pi/4$ at the end, which will mix the amplitudes pair by pair. The probabilities to measure $|e_1\rangle$ and $|e_2\rangle$ are now given by $p(e_1) = (y_1 + y_2)^2$ and $p(e_2) = (y_1 - y_2)^2$. Therefore if $p(e_1) > p(e_2)$, we have $sign(y_1) \neq sign(y_2)$, and if $p(e_1) < p(e_2)$, we have $sign(y_1) = sign(y_2)$. The same holds for the pairs $(y_3, y_4)$, and so on.

    c) We finally perform the same where the *RBS* are shifted by one position below. Then we compare the signs of the pairs $(y_2, y_3)$, $(y_4, y_5)$ and so on.

At the end, we are able to recover each value $y_j$ with its sign, assuming that $y_1 > 0$ for instance. This procedure has the benefit of not adding depth to the original circuit, but requires 3 times more runs. The overall cost of the tomography procedure with sign retrieval is given by $\widetilde{O}(n/\delta^2)$.

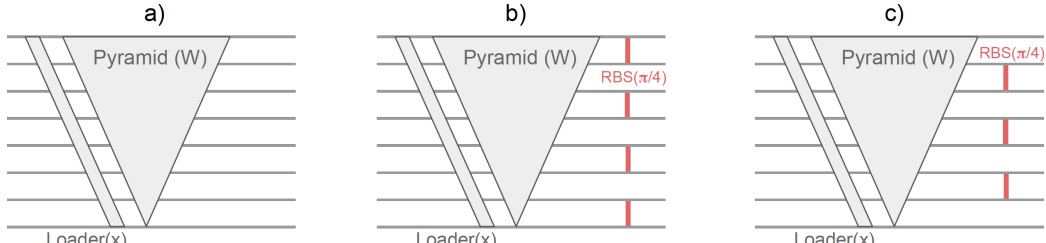

Figure 12: First tomography procedure to retrieve the value and the sign of each component of the resulting vector $|y\rangle = |Wx\rangle$. Circuit a) is the original one while circuits b) and c) have additional *RBS* gates with angle $\pi/4$ at the end to compare the signs between adjacent components. In all three cases an $\ell_\infty$ tomography is applied.

In Fig.13 we propose another method to obtain the values of the amplitudes and their signs. Compared to the above procedure, it relies on one circuit only, but requires an extra qubit and a depth of $3n + O(1)$ instead of $2n + O(1)$.

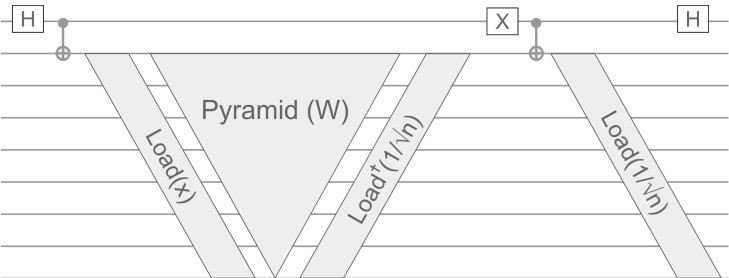

Figure 13: Second tomography procedure to retrieve the value and the sign of each component of the resulting vector $|y\rangle = |Wx\rangle$. For a *rectangular* case with output of size $m$, the two opposite loaders at end must be on the last $m$ qubits only, and the $CNOT$ gate between them connects the top qubits to the loader's top qubit as well.

This circuit initializes the qubits in $(|0\rangle + |1\rangle)|0\rangle$, where the last $|0\rangle$ correspond to the $n$ qubits that will be processed by the pyramidal circuit and the loaders.

Next, applying the data loader for the normalized input vector $x$ (see Section 2.3) and the pyramidal circuit will, according to Eq.(4), map the state to

$$|0\rangle |0\rangle + |1\rangle \sum_{j=1}^{n} W_j x |e_j\rangle \qquad (10)$$

Then, we use an additional data loader for the uniform norm-1 vector $(\frac{1}{\sqrt{n}}, \cdots, \frac{1}{\sqrt{n}})$. Note that this loader is simply built in reverse order to fit the pyramid and limit the augmentation of the depth. We also apply the adjoint of this loader after a controlled operation on the first extra qubit. Recall that if a circuit $U$ is followed by $U^\dagger$, it is equivalent to the identity. Therefore, this will load the uniform state only when the first qubit is in state $|1\rangle$:

$$\frac{1}{\sqrt{2}} |0\rangle \sum_{j=1}^{n} W_j x |e_j\rangle + \frac{1}{\sqrt{2}} |1\rangle \sum_{j=1}^{n} \frac{1}{\sqrt{n}} |e_j\rangle \qquad (11)$$

Finally, a Hadamard gate will mix both parts of the amplitudes on the extra qubit to give us the desired state:

$$\frac{1}{2}\left|0\right\rangle \sum_{j=1}^{n}\left(W_j x + \frac{1}{\sqrt{n}}\right)\left|e_j\right\rangle + \frac{1}{2}\left|1\right\rangle \sum_{j=1}^{n}\left(W_j x - \frac{1}{\sqrt{n}}\right)\left|e_j\right\rangle \tag{12}$$

On this final state, we can see that the difference in the probabilities of measuring the extra qubit in state 0 or 1 and rest in the unary state $e_j$ is given by $\Pr[0, e_j] - \Pr[1, e_j] = \frac{1}{4}\left(W_j x + \frac{1}{\sqrt{n}}\right)^2 - \frac{1}{4}\left(W_j x - \frac{1}{\sqrt{n}}\right)^2 = W_j x/\sqrt{n}$. Therefore, for each $j$, we can deduce the sign of $W_j x$ by looking at the most frequent output of the measurement of the first qubit. To deduce as well the value of $W_j x$, we simply use $\Pr[0, e_j]$ or $\Pr[1, e_j]$ depending on the sign found before. For instance, if $W_j x > 0$ we have $W_j x = 2\sqrt{\Pr[0, e_j]} - \frac{1}{\sqrt{n}}$.

Combining with the $\ell_\infty$ tomography and the non linearity, the overall cost of this tomography is given by $\widetilde{O}(n/\delta^2)$ as well.

### A.4.2 MULTIPLE QUANTUM LAYERS

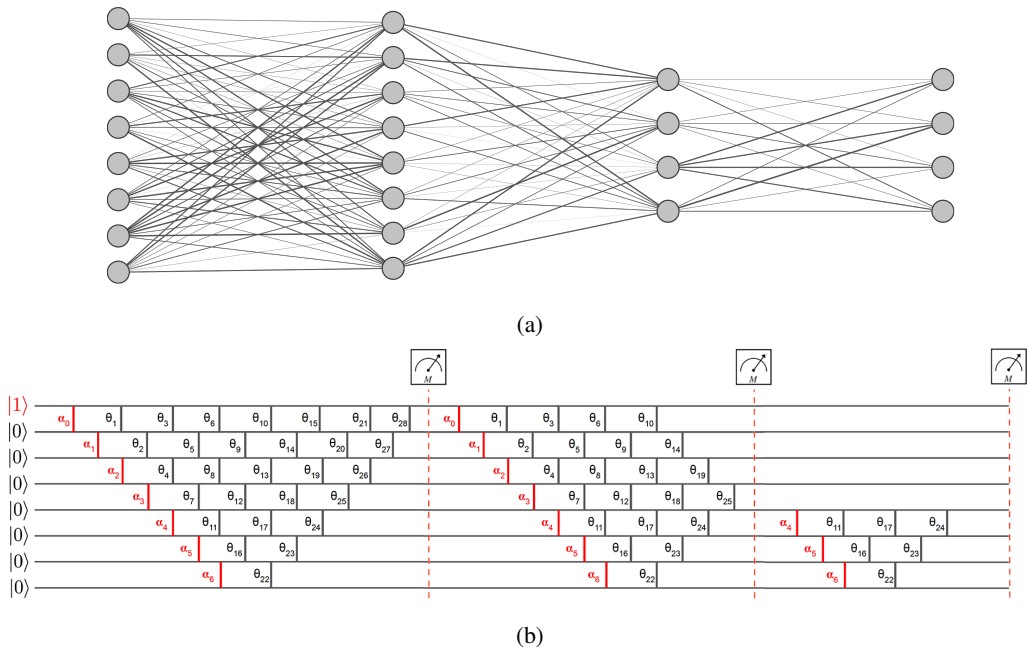

(a)

(b)

Figure 14: A full neural network with layers [8,8,4,4]. (a) Classical representation. (b) The equivalent quantum circuit is a concatenation of multiple pyramidal circuits. Between each layer one performs a measurement and applies a non linearity. Each layer starts with a new unary data loader.

In the previous sections, we have seen how to implement a quantum circuit to perform the evolution of one orthogonal layer. In classical deep learning, such layers are stacked to gain in expressivity and accuracy. Between each layer, a non-linear function is applied to the resulting vector. The presence of these non-linearities is key in the ability of the neural network to learn any function Leshno et al. (1993).

The benefit of using our quantum pyramidal circuit is the ability to simply concatenate them to mimic a multi layer neural network. After each layer, a tomography of the output state $\left|z\right\rangle$ is performed to retrieve each component, corresponding to its quantum amplitudes (see Section A.4.1). A non linear function $\sigma$ is then applied classically to obtain $a = \sigma(z)$. The next layer starts with a

new unary data loader (See Section 2.3). This hybrid scheme allows as well to keep the depth of the quantum circuits reasonable for NISQ devices, by applying the neural network layer by layer.

Note that the quantum circuits proposed in this work can rightfully be called "quantum neural networks" even though this term is usually employed for any arbitrary variational circuit for their closeness to neural networks. With our quantum pyramidal circuits, we control and understand the quantum mapping. It implements each layer and its non linearities, in a modular way. Our orthogonal quantum neural networks are also different regarding the training strategies which are closer to the classical ones (see Section 4 for details). That being said, it is interesting to compare our pyramidal circuit to a variational circuit with $n$ qubits and $n(n-1)/2$ gates of any type, as we usually see in the literature. Using such circuits we would explore among all possible $2^n \times 2^n$ matrices instead of $n \times n$ classical orthogonal matrices, but so far there's no theoretical ground to explain why this should provide an advantage.

Therefore, as an open outlook, one could imagine incorporating additional entangling gates after each pyramid layer (composed, for instance, of $CNOT$ or $CZ$). This would mark a step out of the unary basis but could effectively allow exploring more interactions in the Hilbert Space.

### A.4.3    ADDITIONAL NUMERICAL SIMULATIONS

To complete the results reported in Section 5, we provide additional numerical experiments testing our classical pyramidal circuit for orthogonal neural networks on small use cases, in Fig 15 below.

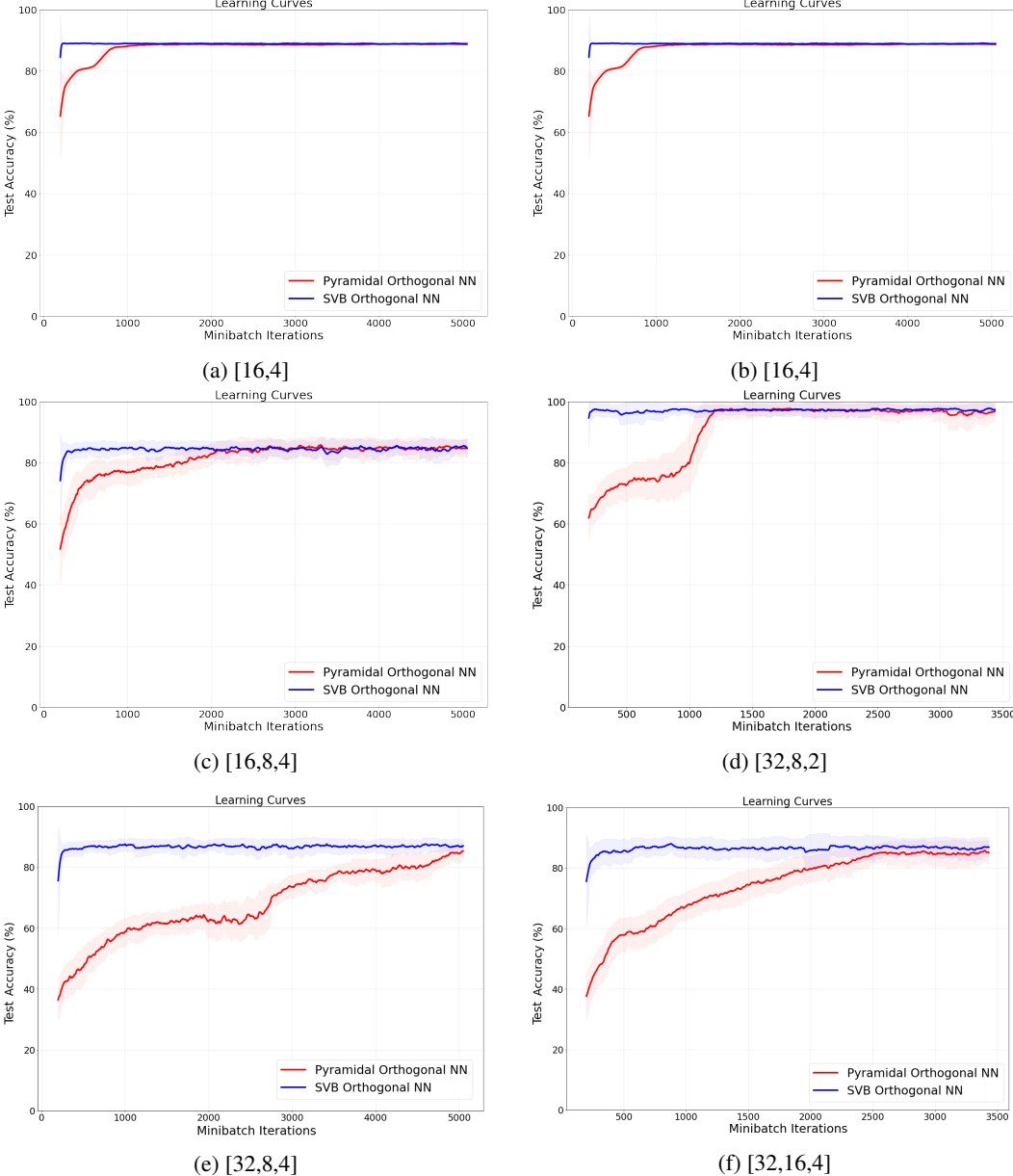

Figure 15: Training comparison between the SVB OrthoNN from Jia et al. (2019) and our classical pyramidal OrthoNN. Test accuracy on 1000 samples during several epochs of training on the MNIST dataset on 5000 samples. Initial dimensionality reduction (PCA) was on the samples to fit the input layer of the networks. Shaded areas indicate the accuracy variance during minibatch updates of size 50.

