# OpenReview forum: "Classical and Quantum Algorithms for Orthogonal Neural Networks"
_ICLR.cc/2022/Conference — ICLR 2022 Submitted_

### Official Review · Reviewer_HBaw · 2021-10-21

**Correctness:** 4
**Technical Novelty And Significance:** 3
**Empirical Novelty And Significance:** 3
**Recommendation:** 6
**Confidence:** 2

**Main Review:**

The novelty of this paper is high. By the introduction of Pyramidal Circuit, the authors can implement an orthogonal matrix multiplication, which allows for gradient descent with perfect orthogonality with the same asymptotic running time as a standard fully connected layer. I think this is a good contribution to the deep learning community. However, the experiments are not very convincing since they lack the comparison of running times.  Overall,  I think the results in this paper are important.


**Summary Of The Paper:**

This paper replaces usual weights and orthogonal matrices in orthogonal neural networks with an equivalent pyramidal circuit made of two-dimensional rotations.  By this, the authors proposed a training method for orthogonal neural networks that run in quadratic time, which is a significant improvement from previous methods based on Singular Value Decomposition. The authors also performed some numerical experiments to verify the abilities of their pyramidal circuit, on the standard MNIST dataset, which showed that their methods are efficient for learning a classification task.

**Summary Of The Review:**

This paper proposed a training method for orthogonal neural networks that run in quadratic time, which is a significant improvement from previous methods based on Singular Value Decomposition. However, the experiments are not very convincing since they lack the comparison of running times.

---

### Official Review · Reviewer_ji2w · 2021-10-26

**Correctness:** 2
**Technical Novelty And Significance:** 2
**Empirical Novelty And Significance:** 2
**Recommendation:** 5
**Confidence:** 2

**Main Review:**

Comments and concerns:

- Reproducibility states "We have explained in detail each part of both algorithms’ implementation, it is in our belief that anyone with classical software skills can re-implement the classical algorithm." If this is the author's belief, what harm would it do to simply release the code? Furthermore, could a classically trained software developer manage to reproduce the experiments on the real quantum device in a reasonable time frame? It is doubtful, as I am a software engineer who works with ibmq devices every day, and I know first-hand that any dealings with such devices are highly nontrivial. I'm doubtful I could reproduce the results in this paper in a reasonable time frame, if at all.

- Section 5 numerical experiments is lacking significant details. The authors claim to achieve 98% accuracy on a binary mnist classification task on the real quantum computer ibmq_bogota. State preparation and measurement error for qubits on this device (very) often exceed 2%. This fact alone makes me highly skeptical of the results. Perhaps the authors did something more clever than what was presented in the paper to achieve this result, but its impossible to know without the code released and without a very detailed description.

- Its unclear to me how significant the results in table 2 are. MNIST with all 10 classes can be classified with >99% accuracy. PCA on MNIST with 2 principle components renders the classes 6 and 9 nearly linearly-separable. I question if 95% accuracy can be achieved by the correct placement of a hyperplane (using 4 principle components). If so, is the network playing a significant role at all?

- Figure 10, if I'm reading this figure correctly, I believe its showing a test accuracy on MNIST of 85%. This is very low compared to state of the art, and low compared to simple off-the-shelf methods. Perhaps I'm not understanding the significance of this, or reading it wrong altogether.

- Section 4 would benefit greatly by being organized into a mathematical proof format, where things are clearly defined beforehand and not just italicized if they mean something particular. Its hard to access the correctness of the claims in the format they're provided in.

- Section 3 states "Therefore, for any given neural network’s orthogonal layer, there is a quantum pyramidal circuit that reproduces it". Again, a statement like this is big, and requires a mathematical proof. Some loose form of justification is provided.

- Equation 5 doesn't make sense to me. If W is an n by n matrix, x is n-dimensional, then Wx is n-dimensional. If U_W is 2^n by 2^n, then how can it operate on x?

- Figure 6, very hard to follow where this matrix came from. I believe its the product of RBS gates.

- Section 2.2 states "The important property to note is that the number of parameters of the quantum pyramidal circuit corresponding to a neural network layer of size n × d is (2n − 1 − d) ∗ d/2 exactly the same as the number of degrees of freedom of an orthogonal matrix of dimension n × d." I believe the authors intended to say "weight matrix" instead of "layer", as a neural network layer can consist of more than a weight matrix.

Typesetting and typos:

- Fig.1. vs Eq.(1). doesn't feel consistent. Furthermore, the absence of a space for "Fig.#" is quite strange. Same with equations, like "n=4" with no spaces.

- Fig. 2(b), the lines in the figure are not a consistent weight and I can not figure out what the intended meaning of this is. If it relates to Fig. 2(a), its unclear how. If it doesn't, the caption isn't sufficient to understand why the network edges are different weights.

- Quotation marks shift in style throughout the paper.

- Suggestion: Instead of repeating multiple times that s(theta) and c(theta) are sine and cosine respectively, state this a single time early in the paper.

- Decimal formatting, virtually all ICLR papers abide by the format "123.50" rather than "123,50" to indicate 123 and one half. This paper uses the latter.

- Figure 5 "7th unary state to the 6th", but mathematically indexing starts from 0, so the 7th unary state corresponds to j=6. This point could potentially confuse.


**Summary Of The Paper:**

The authors propose a new type of neural network layer called a Pyramidal Circuit. This layer implements an orthogonal matrix multiplication, and its claimed that it allows for gradient descent to be run while maintaining perfect orthogonality, and with the same asymptotic running time as an arbitrary fully connected layer. This algorithm is inspired by quantum computing, and it can be applied on a classical computer or a near term (NISQ) quantum computer. To add supporting evidence that the Pyramidal Circuit works, the authors include some experiments demonstrating that networks with Pyramidal Circuit(s) can achieve some level of accuracy on binary classification problems.

**Summary Of The Review:**

The overall idea of pyramidal circuits as neural network layers may have some novel significance. My overall understanding of the evidence supporting the claims in this paper is weak. It is missing any type of demonstration/concrete-proof that network weight matrices remain orthogonal during training, in addition to missing any demonstration of the claimed convergence rate proof. Despite my greater confidence in understanding implementations, I believe its reproducibility is questionable, and the experimental descriptions are lacking. The experimental results appear to be weak, such as achieving an 85% test accuracy on MNIST. Overall it was a difficult paper to access.

---

### Official Review · Reviewer_oStG · 2021-11-02

**Correctness:** 4
**Technical Novelty And Significance:** 3
**Empirical Novelty And Significance:** 3
**Recommendation:** 6
**Confidence:** 4

**Main Review:**

Strengths:
- single, unified architecture that can be implemented on classical and quantum devices
- simple data loading procedure for quantum computer that does not require assumptions about qRAM
- good error characteristics of the unary representation
- experimental results showing similar accuracy for the classical implementation, implementation on a quantum simulator, and implementation on a quantum computer

Weaknesses:
- as the authors note, computing $|Wx>$ for $x$ with $n$ features uses $U_W |x>$ where $U_W$ is a $2^n \times 2^n$ unitary, i.e., one needs $n$ qubits for an $n$-dimensional dataset
- literature review for the classical orthogonal NNs is missing some prior work similar to the pyramidal circuit. Refs. [1] and [2] propose a decomposition of an orthogonal matrix in NNs into a product of simpler matrices, with $O(n^2)$ trainable parameters in total, in a way similar to how Q matrix is decomposed in QR decomposition. [1] uses Householder projections while [2] uses Givens rotations, essentially the same building block as the RBS gate.

[1] Dorobantu et al., DizzyRNN: Reparameterizing Recurrent Neural Networks for Norm-Preserving Backpropagation

[2] Mhammedi et al., Efficient Orthogonal Parametrisation of Recurrent Neural Networks Using Householder Reflections

**Summary Of The Paper:**

A parameterization of NNs that guarantees orthogonality, and is easily implementable using classical as well as quantum computing. The paper proposes a "pyramidal circuit" as a model/architecture that can be implemented on quantum devices and, in the same form, on classical devices. On quantum devices, it is implemented using 2-qubit, 2-level gates (RBS gates), and on classical devices as a sequence of planar rotations. In both cases, there are $n(n-1)/2$ trainable parameters that also only need $O(n^2)$ in the backward pass.

**Summary Of The Review:**

While the efficient parameterization guaranteeing orthogonality is not entirely novel in classical ML, showing a single architecture that works on GPUs and on NISQ devices can gather significant interest from ML and QML communities.

---

### Official Review · Reviewer_h1gX · 2021-11-02

**Correctness:** 3
**Technical Novelty And Significance:** 1
**Empirical Novelty And Significance:** 2
**Recommendation:** 1
**Confidence:** 5

**Main Review:**

Strength:
1. The paper gives a detailed derivation of the construction of the orthogonal NN layer and gives detailed gradient computation.
2. The orthogonal quantum NNs are implemented on real quantum machines.

Weakness:
1. Major: The contribution and novelty of the paper are debatable. Using 2-dimensional unitary planar rotators to construct arbitrary unitary operations is well-established [1]. The beam splitter gate is a tunable Mach-Zehnder interferometer (MZI).
The unitary group parametrization was described in Section 2 of “The Unitary and Rotation Groups, Francis D. Murnaghan, 1962”. Using triangular array (Reck style [2]) and rectangular array (Clements style [3]) to construct arbitrary unitaries was proposed years ago and are widely applied in optical neural networks [4]. Training rotation phases in the unitary parametrization space with various robustness/discretization considerations already exist in the literature [5]. There are also techniques proposed to train rotation phases directly on-chip/in situ without software-assisted backpropagation [6]. Prior work also exists to solve gradient explosion issues in RNNs using those unitary operators [7].

2. Major: The robustness of the constructed orthogonal layer is debatable. The triangular array constructs an arbitrary unitary with 2N-3 depth. Such a deeply-cascaded layer will encounter severe angle error accumulation issues, which is a well-observed phenomenon in optical neural networks [8]. Shallower quantum neural layers might be more resilient to quantum noises. Rectangular (Clements’ style) and butterfly meshes will be more noise-tolerant.

3. Major: Experiments are weak. Not enough data and experiments to support the effectiveness of the orthogonal QNN layer. Discussion on efficiency and robustness of the orthogonal layer is missing.

4. Minor: Training such triangular quantum unitary layers using backpropagation consumes considerable memory as each stage needs to store intermediate results, which is not quite scalable or efficient. Also, training such unitary meshes in the rotation angle space is sensitive to initialization. Without careful initialization, the signals will collapse into the local paths with slow convergence [9].

[1] Nicholas C. Harris, Jacques Carolan, Darius Bunandar, et al., “Linear programmable nanophotonic processors,” Optica, 2018.

[2] M. Reck, A. Zeilinger, H. Bernstein, et al., “Experimental realization of any discrete unitary operator,” Physical review letters, 1994.

[3] William R. Clements, Peter C. Humphreys, Benjamin J. Metcalf, W. Steven Kolthammer, and Ian A. Walmsley, “Optimal design for universal multiport interferometers”, Optica, 2016.

[4] Y. Shen, N. C. Harris, S. Skirloet al., “Deep learning with coherent nanophotonic circuits,” Nature Photonics, 2017.

[5] J. Gu, Z. Zhao, C. Feng, H. Zhu, R. T. Chen and D. Z. Pan, "ROQ: A Noise-Aware Quantization Scheme Towards Robust Optical Neural Networks with Low-bit Controls," DATE, 2020.

[6] Tyler W. Hughes, Momchil Minkov, Yu Shi, Shanhui Fan, “Training of photonic neural networks through in situ backpropagation and gradient measurement,” Optica, 2018.

[7] Li Jing, Yichen Shen, Tena Dubcek, John Peurifoy, et al., “Tunable Efficient Unitary Neural Networks (EUNN) and their application to RNNs,” ICML, 2017.

[8] Michael Y.-S. Fang, Sasikanth Manipatruni, Casimir Wierzynski, et al., “Design of optical neural networks with component imprecisions,” Optics Express, 2019.

[9] Sunil Pai, Ben Bartlett, Olav Solgaard, David A. B. Miller, “Matrix Optimization on Universal Unitary Photonic Devices,” Physical Review Applied, 2019.


**Summary Of The Paper:**

The authors introduce a quantum pyramidal circuit to achieve an orthogonal layer of a neural network, which is fast and can maintain orthogonality. The angle gradients are derived. The authors implement the orthogonal NN on simulators and quantum machines to demonstrate the effectiveness.

**Summary Of The Review:**

The originality of the paper content needs further justification. The method introduced were well-established in photonic computing and the machine learning community.

---

### Decision · Program_Chairs · 2022-01-20

**Decision:**

Reject

**Comment:**

This paper introduces a quantum pyramidal circuit for the computation of orthogonal layers in neural networks and implements the algorithm on simulators and on a quantum computer to illustrate its effectiveness. It also obtains an O(n^2) classical algorithm for forward and backpropagation.

The reviewers generally found strength in the derivations and implementation on real quantum machines. Some reviewers regarded the contributions as strong and novel, while others expressed skepticism about the novelty and the robustness of the algorithm. Having read the paper in detail, I concur with the several reviewers who found the literature review of classical orthogonal NNs to be lacking. In particular, one reviewer highlights similarities with Householder reflections and Givens rotations, for which substantial literature already exists. Without a proper comparison to this existing work, it is not possible to properly assess the novelty or relative contributions of the current paper.

Beyond an extended discussion of related work, the paper would also benefit from improved experimental analysis. While the paper is framed around the quantum algorithm, the main contributions are described as a novel and efficient classical algorithm. This would indeed be a contribution of interest to the broader (non-quantum) ICLR community, but there is no experimental evidence supporting the utility of the proposed methods. An analysis that compares the classical algorithm to the numerous prior works that parameterize orthogonal layers would be an essential addition. As it stands, I cannot recommend the paper for publication.